# Crystal Structure of a Classical MHC Class I Molecule in Dogs; Comparison of DLA-88*0 and DLA-88*5 Category Molecules

**DOI:** 10.3390/cells12071097

**Published:** 2023-04-06

**Authors:** Yujiao Sun, Lizhen Ma, Shen Li, Yawen Wang, Ruiqi Xiao, Junqi Yang, Johannes M. Dijkstra, Chun Xia

**Affiliations:** 1Yantai Institute of China Agricultural University, No. 2006, Binhai Mid-Rd, High-Tech Zone, Yantai City 264003, China; 2Department of Microbiology and Immunology, College of Veterinary Medicine, China Agricultural University, Beijing 100193, China; 3Beijing Institute of Radiation Medicine, 27 Taiping Road, Beijing 100850, China; 4Center for Medical Science, Fujita Health University, Toyoake, Aichi 470-1192, Japan

**Keywords:** crystal structure, dog, MHC, DLA-88, allele, peptide, virus, transmissible cancer

## Abstract

*DLA-88* is a classical major histocompatibility complex (MHC) class I gene in dogs, and allelic DLA-88 molecules have been divided into two categories named “DLA-88*0” and “DLA-88*5.” The defining difference between the two categories concerns an LQW motif in the α2 domain helical region of the DLA-88*5 molecules that includes the insertion of an extra amino acid compared to MHC class I consensus length. We here show that this motif has been exchanged by recombination between different DLA-88 evolutionary lineages. Previously, with pDLA-88*508:01, the structure of a molecule of the DLA-88*5 category was elucidated. The present study is the first to elucidate a structure, using X-ray crystallography, of the DLA-88*0 category, namely DLA-88*001:04 complexed with β2m and a nonamer peptide derived from canine distemper virus (CDV). The LQW motif that distinguishes DLA-88*5 from DLA-88*0 causes a shallower peptide binding groove (PBG) and a leucine exposed at the top of the α2 domain helix expected to affect T cell selection. Peptide ligand amino acid substitution and pMHC-I complex formation and stability analyses revealed that P2 and P3 are the major anchor residue positions for binding to DLA-88*001:04. We speculate that the distribution pattern of the LQW motif among canine classical MHC class I alleles represents a strategy to enhance allogeneic rejection by T cells of transmissible cancers such as canine transmissible venereal tumor (CTVT).

## 1. Introduction

Canines have a unique set of characteristics that meet human needs. At least fifteen thousand years ago, humans domesticated wolves that evolved into today’s dogs [1,2]. This has resulted in approximately 400 breeds and hundreds of millions of dogs. Dogs are the most successful animals to enter the human family and play various important roles in human society.

As in other mammals, the dog immune system confers immune memory, and this allows vaccination. To date, the development and application of vaccines for viral canine diseases, such as canine distemper virus (CDV), rabies virus, canine parvovirus (CPV), canine adenovirus (CAV), and canine coronavirus (CCV), have been successful [3,4,5,6,7]. However, because for some pathogens vaccines have not been developed yet—or need improvement—and new infectious diseases keep emerging [8], a better understanding of the adaptive immune system in dogs should be obtained. That will not only strengthen the prevention and control of viral diseases within dogs but also their possible transmission from dogs to other species, like humans, such as in the case of the rabies virus.

Similarly to other mammals, dogs have B cells that make antibodies and CD4^+^ and CD8^+^ T cells that presumably screen antigens presented by major histocompatibility complex (MHC) class II and I (MHC-II and MHC-I) molecules, respectively [9,10,11]. In the canine *Mhc* genomic region on Chr. 12—named the *DLA* (*dog leukocyte antigen*) complex—MHC-I and MHC-II genes are found as in the *Mhc* of other mammals [12,13]. DLA matching between effector and target cells was found to promote killing by cytotoxic lymphocytes [14], suggesting classical MHC-I restriction. There are three MHC-I genes in the *DLA* complex, named *DLA-88*, *DLA-88L/DLA-12*, and *DLA-64* [12]. There is also an additional MHC-I gene on chromosome 18, named *DLA-79* [15]. Among these four genes, *DLA-88* is a typical classical MHC-I gene because of its classical type polymorphism, ubiquitous expression, and the encoding of the characteristic “key” amino acids for peptide ligand binding, while *DLA-64* and *DLA-79* can be classified as nonclassical [15,16]. Probably as the result of recombination, the alleles of the *DLA-88L/DLA-12* locus belong to two different lineages, *DLA-88-like* (*DLA-88L*) and *DLA-12*, of which the former are very similar to *DLA-88* alleles and are higher expressed than the *DLA-12* type alleles [17,18].

Currently, 140 DLA-88 alleles combined for dog (*Canis lupus familiaris*) and gray wolf (*Canis lupus lupus*) are listed in the MHC section of the Immuno Polymorphism Database (“IPD-MHC,” https://www.ebi.ac.uk/ipd/mhc/, accessed on 10 December 2022). The allele names either start with “DLA-88*0” or “DLA-88*5,” based on an additional amino acid in the α2 domain helical regions in the DLA-88*5 category [19]. Hitherto, the only structural analysis of DLA-88 molecules has been for the allele DLA-88*508:01 [20]. The present study is a first report of a structure, determined by X-ray crystallography, of an allele of the DLA-88*0 category, namely DLA-88*001:04. Amino acid substitution analysis revealed that anchor residues for binding DLA-88*001:04 involve peptide ligand positions P2 (“P” for peptide ligand position) and P3. We discuss a possible function of the LQW motif that sets the DLA-88*0 and DLA-88*5 categories apart.

## 2. Materials and Methods

### 2.1. Analysis of DLA-88 Sequences

Sequences were retrieved from the MHC dataset of the Immuno-Polymorphism Database (IPD) [21] (https://www.ebi.ac.uk/ipd/mhc/, accessed on 10 December 2022), the National Center for Biotechnology Information (NCBI) (https://pubmed.ncbi.nlm.nih.gov, accessed on 10 December 2022), and UniProt (https://www.uniprot.org/, accessed on 10 December 2022). UniProt provided information on whether the sequences were found in dog or gray wolf, and provided links to NCBI accessions with detailed information. For DLA-88 alleles found in wolf, we tried to find their reporting in dogs by BLASTP similarity searches in the NCBI database and by checking the literature. Except for two DLA-88 alleles in CTVT cancer cells, only sequences that were (also) deposited in the IPD database—which we interpreted as a quality mark by the dog MHC research community—were analyzed in the present study. Sequences were aligned by hand [22], which in comparison to computerized alignments corrects and homogenizes the position of the gap in the α2 domain; this position agrees with the structural comparison between pDLA-88*001:04 and pDLA-88*508:01. The evolutionary history was inferred using the Neighbor-Joining method [23]. The bootstrap consensus tree inferred from 500 replicates was taken to represent the evolutionary history of the taxa analyzed [24]. Branches corresponding to partitions reproduced in less than 50% bootstrap replicates were collapsed. The evolutionary distances were computed using the Poisson correction method [25] and are in the units of the number of amino acid substitutions per site. The analysis involved 142 amino acid sequences (140 from the IPD database and two reported for CTVT). All ambiguous positions were removed for each sequence pair. There were a total of 182 positions in the final dataset. Evolutionary analyses were conducted in MEGA7 [26]. Indications of β-strand and helix structures follow structural element assignments in the PDB database of NCBI.

### 2.2. Synthesis of Viral Peptides

A total of 18 different viral nonapeptides were used (Appendix A). These nonapeptides corresponded to sequences of CDV, CPV, rabies virus, and influenza virus that were predicted to have high affinity for DLA-88*001:04 by NetMHCpan-4.0 (accessed on 10 July 2020 http://www.cbs.dtu.dk/services/NetMHCpan), and also included the KLF9 self-peptide that was previously reported to bind DLA-88*508:01 [20]. These peptides were synthesized and purified by reversed-phase high-performance liquid chromatography (SciLight Biotechnology, Beijing, China) with >95% purity. These lyophilized peptides were stored at −20 °C, were dissolved in DMSO at a concentration of 25 mg/mL before use, and then used as described.

### 2.3. Protein Preparation

Gene fragments were synthesized encoding the extracellular domain (1-275 amino acids) of DLA-88*001:04 (GenBank accession NP_001014767) and the mature protein (1-98 amino acids) of canine β2m (GenBank accession No. JQ733515), while adding restriction enzyme target sites for convenient cloning, by Shanghai Invitrogen Life Technologies. The *DLA-88*001:04* and canine *β2m* genes were then, after digestion by NdeI and XhoI enzymes, ligated into the pET21a expression vector (Novagen, Merck KGaA, Darmstadt, Germany) and transformed into *E. coli* BL21 (DE3) cells. Recombinant DLA-88*001:04 was expressed in inclusion bodies and purified as described previously [27]. Recombinant canine β2m was also expressed in inclusion bodies and purified as previously described [6]. Finally, the inclusion bodies of DLA-88*001:04 and canine β2m were dissolved in 6 M guanidine hydrochloride solution (6 M guanidine hydrochloride (Gua-HCl), 50 mM Tris-HCl (pH 8.0), 10 mM EDTA, 100 mM NaCl, 10% *v*/*v* glycerol and 10 mM DTT). The final concentration was 30 mg/mL. The proteins were stored at −20 °C.

### 2.4. The Assembly and Purification of the pDLA-88*001:04 Complex

To form a complex with each peptide, DLA-88*001:04, β2m, and the nonapeptide were refolded at a ratio of 3:1:1 using the gradual dilution method, as previously described [28]. As a negative control, DLA-88*001:04 and β2m were also refolded without the peptide. After 48 h of incubation at 4 °C, the remaining soluble portion of the complex was concentrated and then purified by chromatography on a Superdex 200 16/60 column followed by Resource Q anion-exchange chromatography (GE Healthcare, Chicago, IL, USA), as previously described [29]. Finally, the purified complex protein solution was replaced with 10 mM Tris-HCl and 50 mM NaCl (pH 8.0) three times.

### 2.5. Thermostability Measurements Using Circular Dichroism Spectroscopy

The thermostabilities of DLA-88*001:04 complexed with β2m and different nonapeptides were tested by CD spectroscopy. CD spectra were measured at 20 °C on a Jasco J-810 spectropolarimeter (Oklahoma City, OK, USA) equipped with a temperature-controlled cell holder. The protein concentration was 0.1 mg/mL in Tris buffer (20 mM Tris, 50 mM NaCl, pH 8.0). Thermal denaturation curves were determined by monitoring the CD value at 218 nm using a 1-mm optical-path-length cell as the temperature was raised from 25 to 80 °C at a rate of 1 °C/min. The temperature of the sample solution was directly measured with a thermistor. The fraction of unfolded protein was calculated from the mean residue ellipticity (θ) using a standard method. The unfolded fraction (%) is expressed as (θ − θN)/(θU − θN), where θN and θU are the mean residue ellipticity values in the fully folded and fully unfolded states, respectively. The Tm was determined by fitting the data to the denaturation curves using the Origin 8.0 program (OriginLab Northampton, MA, USA) as described [30].

### 2.6. Crystallization and Data Collection

One viral peptide, CDV-hemagglutinin derived RTI9 (RTISYTYPF), was selected for crystallization with the DLA-88*001:04 H chain and canine β2m. The RTI9/DLA-88*001:04/β2m complex was concentrated to 6 and 12 mg/mL in buffer containing 20 mM Tris (pH 8.0) and 50 mM NaCl for crystallization. After being mixed with reservoir buffer at a 1:1 ratio, the purified complex was crystallized by the hanging-drop vapor diffusion method at 291 K. Polyethylene glycol (PET)/ion kits (Hampton Research, Riverside, CA, USA) were used to screen for crystals. After several days, crystals of the RTI9/DLA-88*001:04/β2m complex were obtained with solution 33 (20% (*w*/*v*) PEG 3350 and 0.2 M sodium sulfate decahydrate) at a concentration of 12 mg/mL. Diffraction data were collected using an in-house X-ray source (Rigaku MicroMax007 desktop rotating anode X-ray generator (Tokyo, Japan) with a Cu target operated at 40 kV and 30 mA) and an R-Axis IV++ imaging plate detector at a wavelength of 1.5418 Å. In each case, the crystal was first soaked in a reservoir solution containing 15% glycerol as a cryoprotectant for several seconds and then flash-cooled in a stream of gaseous nitrogen at 100 K [31]. The collected intensities were indexed, integrated, corrected for absorption, scaled, and merged using HKL2000 [32].

### 2.7. Structure Determination and Refinement

The structure of the RTI9/DLA-88*001:04/β2m complex was solved by molecular replacement using the MOLREP program with SLA-1*0401 (PDB ID: 3 QQ3) as the search model. Extensive model building was performed by hand using Coot [33], and restrained refinement was performed using REFMAC5. Further rounds of refinement were performed using the phenix.refine program implemented in the PHENIX package with isotropic ADP refinement and bulk solvent modeling, which improved the R and Rfree factors from 0.194 and 0.209 to 0.151 and 0.177, respectively. The stereochemical quality of the final model was assessed with the PROCHECK program [34]. Data collection and refinement statistics are listed in Table 1.

### 2.8. Analysis and Figure Depictions of Structural Data

Structural analyses and their figure depictions were performed using PyMOL (https://www.pymol.org, accessed on 10 December 2022), Collaborative Computational Project Number 4 (CCP4) [35], and Coot [33]. Interactions within 4.5 Å distance between peptide RTI9 and DLA-88*001:04 PBG were determined by CCP4.

The coordinates and structure factors of the RTI9/DLA-88*001:04/β2m complex have been deposited in the Protein Data Bank with the accession number 7CJQ.

## 3. Results

### 3.1. DLA-88 Sequences in Dogs and Gray Wolves; the LQW Motif in the α2 Domain That Dsitinguishes Categories DLA-88*0 and DLA-88*5 Is not Restricted to a Single DLA-88 Lineage

The α1 + α2 domain sequences of all the 140 DLA-88 alleles available in the IPD-MHC database were compared. Of these, to the best of our knowledge, 131 were only reported for dog (*Canis lupus familiaris*), six only for gray wolf (*Canis lupus lupus*), and three for both subspecies, but further investigations will probably find that more alleles are shared. A phylogenetic tree is shown in Figure 1, and an alignment of sequences representative of the different branches of the tree while including all DLA-88*5 category sequences is shown in Figure 2 (an alignment figure including 140 DLA-88 sequences can be seen at the IPD-MHC database). In Figure 1 and Figure 2, the DLA-88*5 category sequences are highlighted in yellow, and also in yellow, in Figure 2, is their defining motif, LQW, around position 155 in the α2 domain. This LQW motif introduces an extra residue insertion into the α2 helical region as compared to the MHC-I consensus situation [36], the latter being exemplified in Figure 2 by human HLA-A2 and the canine DLA-88*0 category alleles.

The mosaic distributions of the LQW motif and other sequence motifs, highlighted in Figure 2 by green, pink, and cyan shading, are indicative of interallelic recombination events. The two DLA-88 alleles of canine transmissible venereal tumor (CTVT) [39], here denoted as DLA-88*CTVT-1 and -CTVT-2 (they are not included in the PDB database), do not possess the LQW motif (Figure 1 and Figure 2).

### 3.2. Overall Structure of the pDLA-88*001:04 Complex

Eighteen peptides, which were predicted to bind well to DLA-88*001:04, were synthesized for testing. Among these peptides, twelve contributed to the formation of stable peptide/DLA-88*001:04/β2m complexes after in vitro refolding (Appendix A). Finally, the complex structure of DLA-88*001:04 together with β2m and the nonamer peptide RTI9 (sequence RTISYTYPF) derived from canine distemper virus (CDV) was solved. The X-ray data collection statistics of this structure—which for simplicity is named pDLA-88*001:04 (the “p” refers to being a complex of heavy chain, β2m, and peptide) in most of this study—are summarized in Table 1. The overall structure of pDLA-88*001:04 (Figure 3A) is similar to known pMHC-I structures, and its pMHC-I-characteristic distributions of β-strands and helices in the α1 and α2 domains are highlighted in Figure 2. In pDLA-88*001:04, the polar interactions between heavy (H) chain and β2m include the highly conserved pMHC-I-typical interactions between β2m Trp-59 and H chain Gln-96 and Asp-119, between β2m Asp-52 and H chain Arg-48, and between β2m Tyr-10 and H chain Pro-235 (Figure 3C) [36,40,41].

**Table 1 cells-12-01097-t001:** X-ray data diffraction data processing and refinement statistics.

Parameter or Statistic	RTI9/DLA-88*001:04/β2m
Data processing	
Space group	P212121
Cell parameters (Å)	A = 89.3, b = 93.0, c = 119.4
Resolution range (Å)	50.00–2.70 (2.75–2.70)
Total reflections	152570
Unique reflections	50958
Completeness	0.978 (0.969)
*R*merge	0.201 (0.540)
I/σ	7.308 (2.500)
*R*-value Work	0.253
*R*-value Free	0.282
Bonds (Å)	0.004
Angles (°)	0.926
Average B factor	17.013

*R*merge = Σhkl Σi|Ii (hkl)–〈I (hkl)〉|/Σhkl Σi Ii (hkl), where Ii (hkl) is the observed intensity and 〈I (hkl)〉 is the average intensity from multiple measurements. *R* factor = Σ (Fobs-Fcalc)/ΣFobs; *R*-value Free is the *R* factor for a subset (5%) of reflections that was selected prior to refinement calculations and not included in the refinement.

### 3.3. The Conformation of the Peptide Binding Groove in pDLA-88*001:04

Based on the pockets in the peptide binding groove (PBG) of human pHLA-A2 [40], PBGs of various pMHC-Is have been divided into pockets A-to-F regions, as we do for pDLA-88*001:04 in the present study (Figure 4). However, it should be noted that, like in some other pMHC-Is (e.g., [42]), the pockets C-to-E of pDLA-88*001:04 might better be understood as a large central cavity than as three separate cavities. The interactions between RTI9 peptide and the PBG are listed in Table 2.

As common among pMHC-I structures, the sidechain of the RTI9 peptide residue at position 1 (P1) points upwards and the main chain of the P1 residue participates in a hydrogen bond network that also includes Tyr-7, Tyr-59, Glu-63, Tyr-159, Tyr-171, and a water molecule (Figure 4, pocket A) [36,41].

In most known pMHC-I structures, the B pocket is an important anchor site that harbors the sidechain of the peptide ligand P2 residue and plays a selective role in peptide binding [43,44]; also, in pDLA-88*001:04 the side chain of P2-Thr inserts into the B pocket (Figure 4). P2-Thr is bound by polar contacts with sidechains of Glu-63 and Thr-66, and the one between the P2 main chain and Glu-63 (or Gln-63) is quite common among pMHC-Is [36]. The B pocket in pDLA-88*001:04 is not very big and not highly charged, which probably explains the acceptance at the peptide ligand P2 position of residues Ala, Asn, Leu, Pro, Ser, Thr, Val, and Pro (Appendix A).

In the C pocket of pDLA-88*001:04, the sidechain of P6-Thr makes polar contacts with the sidechain of Arg-73 (Figure 4).

Reminiscent of many other pMHC-Is [36], the sidechain of P3-Ile inserts into the D pocket (Figure 4). The main chain of P3-Ile forms a polar contact with the sidechain of Tyr-99. The D pocket of pDLA-88*001:04 is relatively hydrophobic, which may explain why stable complexes with peptide ligands carrying P3 residues Ala, Asn, Ile, Leu, Met, Phe, Thr, and Val were found (Appendix A).

The E pocket of pDLA-88*001:04 is quite large. Binding of the RTI9 peptide includes hydrogen bonds between the Trp-147 sidechain and the P7 and P8 main chains (Figure 4 and Appendix A), of which the latter hydrogen bond is quite well conserved among pMHC-Is [36,41]. We speculate that, compared to RTI9 peptide, other peptides bound to DLA-88*001:04 may have their P7 sidechain inserted more downwards into the E pocket because of the available space.

In the F pocket of pDLA-88*001:04, reminiscent of other pMHC-I structures [36,40,41,43], the sidechain of the C-terminal peptide ligand residue, P9-Phe in this case, inserts downwards into the F pocket and its main chain makes polar contacts with the sidechains of Asp-77, Tyr-84, and Thr-143 (Figure 4). The F pocket of pDLA-88*001:04 is large and hydrophobic as well as slightly negatively charged (Figure 4), which may explain the tolerance for the P9 residues Arg, Ile, Leu, Lys, Met, and Phe (Appendix A).

Appendix A compares the folding of the RTI9 peptide main chain in the pDLA-88*001:04 structure with those of peptides in several other representative pMHC-Is, revealing the “M” conformation and an elevation of the P4-to-P8 stretch that are quite commonly found among peptide ligands in pMHC-Is [36].

### 3.4. “Bimodal” Structural Motif Differences between DLA-88 Alleles, Exemplified by pDLA-88*001:04 and pDLA-88*508:01 Structures

When comparing elucidated pDLA-88*001:04 and pDLA-88*508:01 [20] structures, there appear to be two structural motifs that seem to represent a rather bimodal variation among known DLA-88 alleles in both wolf and dog and probably have a significant functional impact.

In many DLA-88 alleles, amongst which DLA-88*508:01, the residue at position 114 is much bigger (Arg-114 or Trp-114) than Ser114 found in pDLA-88*001:04 and about half of the other DLA-88 alleles (Figure 2 and see the IPD-MHC database for all known alleles); among the 142 DLA-88 sequences compared in the present study, 67 have Ser-114, 1 has Asp-114, 58 have Arg-114, and 16 have Trp-114. The absence of a large residue at this position like the arginine in pDLA-88*508:01 makes the E pocket of DLA-88*001:04 considerably larger (Figure 5).

An even more interesting difference is the LQW motif in the α2 domain which gave rise to the nomenclature distinction into the DLA-88*0 and DLA-88*5 categories. The Leu-154a residue is exposed at the top of the helix (Figure 6). In the peptide binding groove, the bulky Trp-156 creates a separation between the D and the E pockets, and Xiao et al., 2016 [20], described Trp-156 as part of an orthogonal wall of the D pocket.

### 3.5. Amino Acid Substitution Analysis Reveals Anchor Residue Properties at Peptide Ligand Positions P2 and P3

To analyze the primary anchor positions for binding DLA-88*001:04, the RTI9 peptide and residue-substituted variants thereof were tested for pDLA-88*001:04 complex formation and stability after in vitro refolding by gel filtration and circular dichroism spectroscopy experiments (Figure 7). The results show that the midpoint transition temperatures (Tm) of the pDLA-88*001:04 complexes were most decreased after P2-(Thr-to-Phe) or P3-(Ile-to-Ala) substitutions, followed by P5-(Tyr-to-Ala). On the other hand, an increased stability was observed after a P2-(Thr-to-Val) substitution. The data reveal P2 as an important anchor position, and suggest that the residue should not be too big and that a hydrophobic residue is preferred over a hydrophilic one; this is consistent with the pocket structural data (Figure 4). The results also indicate P3 as an anchor position, especially because the P3-Ile sidechain of RTI9 in the pDLA-88*001:04 complex seems not to engage in special stabilizing intrapeptide interactions that otherwise might explain its selection (Figure 4); on the other hand, the results of the complex building experiments summarized in Appendix A indicate that at the P3 position a variety of non-charged residues are tolerated. The P5 residue of RTI9 in pDLA-88*001:04 does not insert into a pocket, but we assume that a tyrosine is preferred here over an alanine because of intra-peptide stabilization between P5-Tyr and P7-Tyr (Figure 4). The P9 position appears to not be very selective, although P9-Phe is slightly preferred over P9-Ala (Figure 7); that the P9 position in DLA-88*001:04 is not very selective agrees with the complex building experiments summarized in Appendix A.

## 4. Discussion

The alleles of the canine classical MHC-I molecule DLA-88 have been classified into DLA-88*0 and DLA-88*5 categories based on an extra residue in the α2 domain [19] which is part of an LQW motif. The present study shows that both the presence and absence of this motif are found in multiple different phylogenetic DLA-88 lineages (Figure 1 and Figure 2), which is indicative of past interallelic recombination events and a strong balancing selection. Evidence of interallelic *DLA-88* recombination events has been presented before [18,38] and interallelic recombination events have also been concluded for MHC-I in species other than canines (e.g., [45,46]). Why the residues Leu-154a, Gln-155, and Trp-156 seem to be distributed as a set is unclear, but they probably affect each other’s conformation.

The present study is the first to determine a pMHC-I structure of the DLA-88*0 category molecules, allowing the comparison of a structure without the LQW motif, namely pDLA-88*001:04, with the previously reported pDLA-88*508:01 structures that do possess the motif [20]. Figure 6 shows that one of the pronounced differences made by this motif is the exposure of Leu-154a at the top of the α2 helix, which is at an elevated position seemingly because of the extra bulkiness by having an extra amino acid. Xiao et al., 2016 [20], already realized that Leu-154a is exposed at the surface in a known TCR docking region. Indeed, the TCR binding to pMHC-I complexes often involves direct contacts with this region of the H chain [47,48].

The allelic variation in having and not having an elevated and exposed Leu-154a is not common among MHC-I in other species—we do not know of another example—and we speculate that the differences in T cell clones selected may help to more rigorously reject transferrable tumors that carry MHC-I alleles without such a motif. Such more vigorous rejection may help to block new transmissible tumors from establishing and limit the spread of the canine transmissible venereal tumor (CTVT) cells that infect both dogs and wolves [49,50]. CTVT cells are transferred via sexual intercourse and have alleles without the LQW motif (Figure 2) [37]. Rejection of allogeneic cells by T cells depends on T cell cross-reaction with non-identical MHC molecules [51], and—although it is not known which type of MHC-I differences cause the strongest allogeneic rejection—it can be speculated that the possession of an allele with an LQW motif increases the number of T cell clones that show a strong reaction with CTVT cells. In agreement with such a model, the oral spread of a transmissible cancer in Tasmanian devils is believed to be related to low levels of MHC diversity [52]. Balancing selection for maintaining MHC allelic variation is generally considered to be driven by different alleles presenting different peptides, thereby increasing the resistance of a population to pathogens that otherwise by mutating might more easily escape the immune protection in all individuals of a population (e.g., [53]). However, in teleost fish the MHC class I sequence information indicates that this cannot be the only reason, because in those species the conserved ancient allelic variation is not restricted to the PBG, and a function in allograft rejection has been proposed [54]. It was hypothesized that because of their mucosal skin and living in water, in teleost fish the allograft rejection against potential transferable cancers should be especially important [54], and the DLA-88*0 versus DLA-88*5 differences in canines may be an example of how allelic MHC variation can also be increased in mammals, to help allograft rejection of cancers that are transferred through mucosal tissues.

The Trp-156 residue of the LQW motif makes a wall in the D pocket so that in comparison to pDLA-88*001:04 the D pocket is no longer continuous with the E pocket (Figure 6). Another “bimodal” variation among DLA-88 alleles with a large impact on the groove is that having or not having of a large residue at position 114 that can make the E pocket much smaller (Figure 5). Having “bimodal” types of allelic variations is not unique for canine DLA-88 (e.g., see the various alignments of MHC-I alleles in the IPD MHC database), but most studies do not discuss this. We speculate that this type of variation is driven by simultaneous requirements for sufficient structural stability of the molecule and a need for groove variation related to pathogen resistance. It is unclear whether the Trp-156 residue in the LQW motif is selected for its groove properties or to help fixate the Leu-154a residue in a proper position for affecting TCR docking.

Amino acid substitution analysis (Figure 7) revealed that the main anchor residues for peptide ligand binding to DLA-88*001:04 are P2 and P3, which have their sidechains inserted into the B and D pockets, respectively (Figure 4). In different animals, P2 and the C-terminal residue (PΩ) are the anchor positions most commonly found among pMHC-I complexes (e.g., [36,44]). For DLA-88*001:04, there appeared to be only a weak selection for the PΩ residue (Figure 7 and Appendix A). P2 and PΩ were also—but not in all cases the only—anchor positions found for dog DLA-88*501:001 [55], DLA-88*508:01 [20,56], and DLA-88*034:01 [57].

By pMHC-I complex refolding experiments we checked which peptides bound stably to DLA-88*001:04. This revealed that most of the peptides predicted by NetMHCpan-4.0 software did bind well indeed (Appendix A). Hence, this type of prediction may be used as an initial selection for peptides that might be used as peptide epitope vaccines in dogs. We hope that the peptides of rabies, influenza, and CDV viruses that we found to bind DLA-88*001:04 may contribute to vaccine development.

The present study reveals structural characteristics of the MHC-I system in dogs, in particular about the differences between DLA-88*0 and DLA-88*5 category molecules. More in general, it highlights that MHC-I allelic evolution can involve the exchange between allelic lineages of sequence motifs that have a large structural effect on groove properties and T cell binding sites. The study also identifies virus-derived peptides that may help the development of future peptide epitope vaccines for dogs.

## Figures and Tables

**Figure 1 cells-12-01097-f001:**
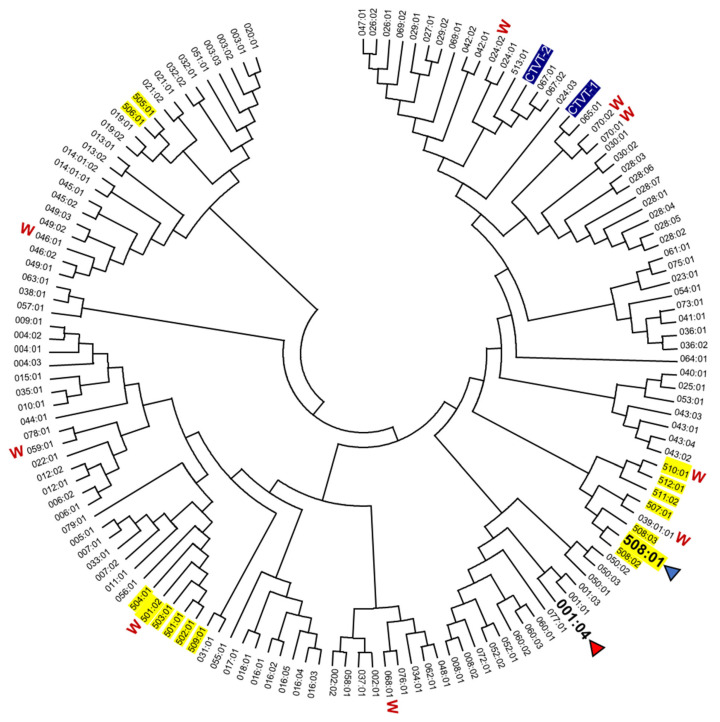
Phylogenetic relationship among DLA-88 alleles. The phylogeny of the α1 + α2 domains was estimated by Neighbor-Joining method for all DLA-88 alleles of dog and gray wolf (W) deposited in the IPD-MHC database. DLA-88 alleles for which the structure has been determined are highlighted with an arrowhead. DLA-88 alleles of the DLA-88*5 category are highlighted with yellow. For GenBank accession numbers of most DLA-88 alleles see the IPD-MHC database. Other GenBank accession numbers are: DLA-88*CTVT-1, AH015035; DLA-88*CTVT-2, AH015036. A few of the here compared sequences of the DLA-88*0 category may be expressed from the *DLA-88L*/*DLA-12* locus [37]. The DLA-88 alleles *024:02 ([37] and GenBank LR890075), *039:01 [37,38], and *046:01 (GenBank AKS10581 and [38]) were reported for both dog and gray wolf.

**Figure 2 cells-12-01097-f002:**
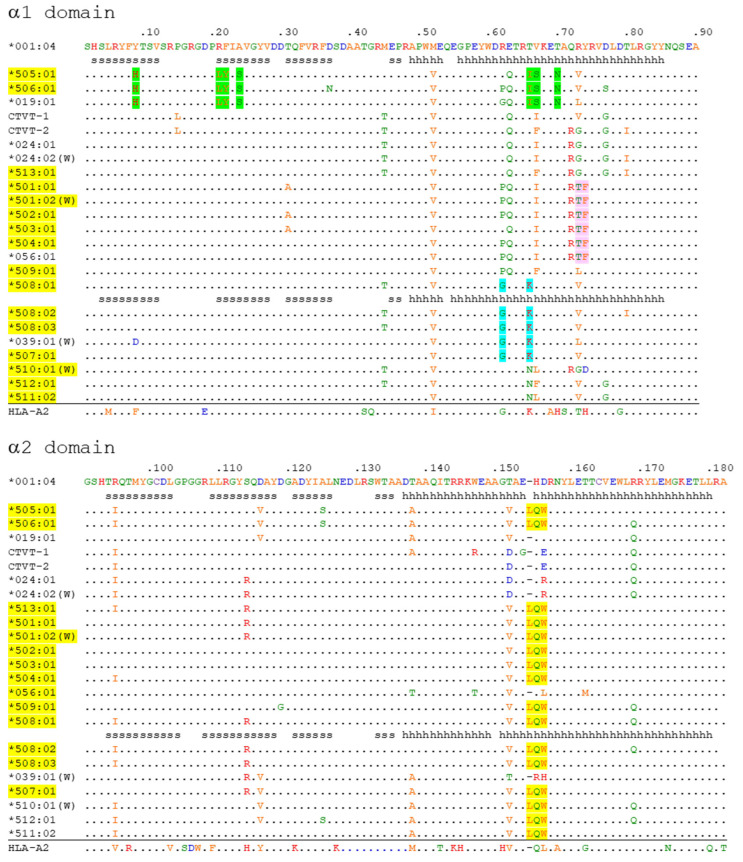
Alignment of deduced DLA-88 amino acid α1 + α2 domain sequences. Sequences representative for different branches of the DLA-88 phylogenetic tree (see Figure 1) and all known alleles of the DLA-88*5 category (yellow) are compared together with human HLA-A2. The DLA-88*5 category motif LQW is highlighted in yellow, and three other motifs are highlighted in green, pink, and cyan; the distribution pattern of LQW and the other highlighted motifs among the alleles is indicative of past interallelic recombination events. W, (also in) wolf; periods, identical residues; hyphens, gaps introduced in the alignment; s and h, beta-strands and helices, respectively, according to PDB accession 7CJQ for DLA-88*001:04 and PDB 5F1I for DLA-88*508:01. For GenBank accession numbers of most DLA-88 alleles see the IPD-MHC database. Other GenBank accession numbers are: DLA-88*CTVT-1, AH015035; DLA-88*CTVT-2, AH015036; HLA-A2, CAB4529282.

**Figure 3 cells-12-01097-f003:**
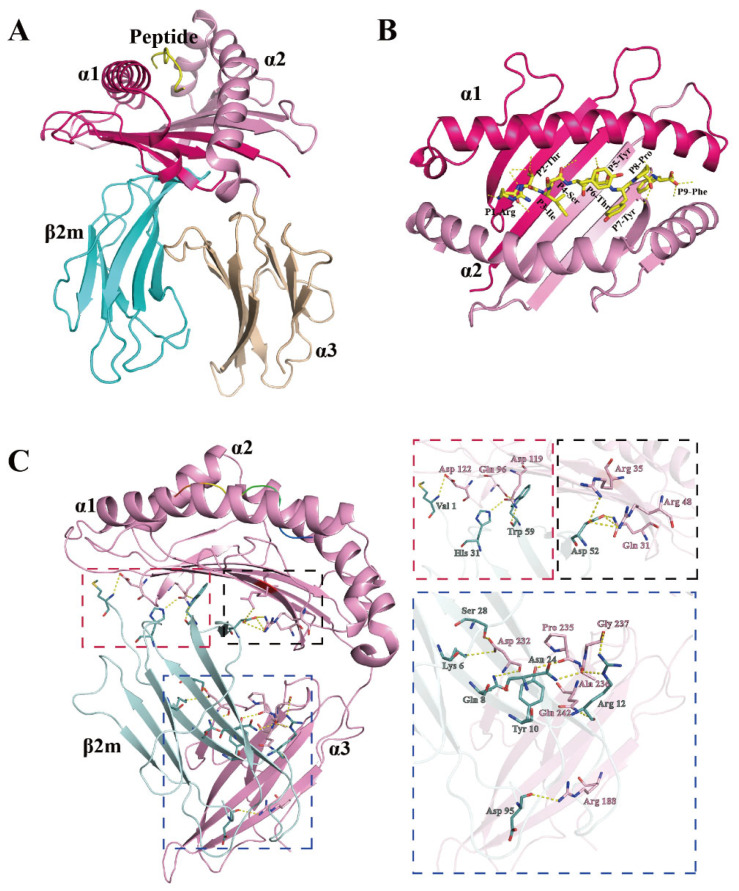
Structure of the dog heterotrimeric pMHC-I complex RTI9/DLA-88*001:04/β2m (pDLA-88*001:04). (**A**) Frontal view of the whole complex in cartoon format. The domains of the H chain are shown in hot pink (α1 domain), light pink (α2 domain), and beige (α3 domain), while β2m is in cyan and the CDV-derived peptide RTI9 (RTISYTYPF) is in yellow. (**B**) Top view of the α1 and α2 domains; the RTI9 peptide is shown in sticks format and its polar interactions with the H chain are indicated with dashed yellow lines. (**C**) Polar interactions between the H chain and β2m. The H chain is shown in pink and β2m in teal. Highlighted residues are shown in sticks format with element-coloring (blue, nitrogen; red, oxygen). Polar contacts (hydrogen bonds) are shown as yellow dashed lines. Interdomain interaction sites are boxed with dotted lines, and shown individually and enlarged at the right in boxes of matching color.

**Figure 4 cells-12-01097-f004:**
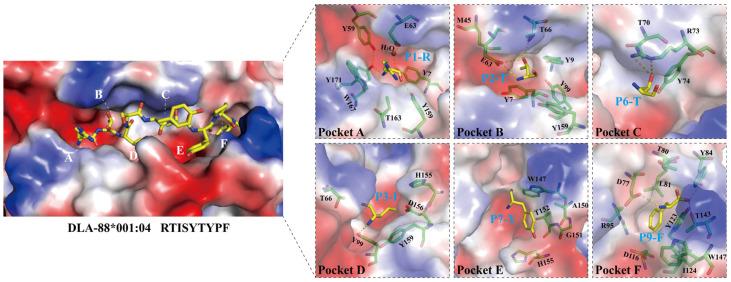
Structural analysis of the PBG of pDLA-88*001:04, its peptide ligand RTI9, and its six binding pockets A-to-F. The H chain is shown in surface format and colored based on vacuum electrostatics: red represents negatively charged, blue represents positively charged, and white represents noncharged. The RTI9 residues are shown in sticks format and element coloring; yellow for carbon, red for oxygen, blue for nitrogen. In the figures on the right, the individual pockets are enlarged and the surface presentation of the H chain is transparent; the H chain residues that interact with the indicated RTI9 residue are highlighted in magenta sticks format.

**Figure 5 cells-12-01097-f005:**
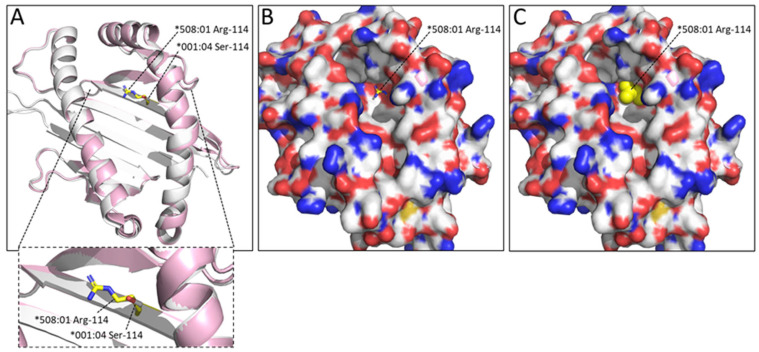
The Ser-114 residue in pDLA-88*001:04 leaves considerably more space in the E pocket than the Arg-114 residue found in pDLA-88*508:01 (PDB accession 5F1I). Superposition of α1α2 domains: ((**A**), with zoomed-in view) pDLA-88*001:04 in white and pDLA-88*508:01 in pink, with the sidechains of their Ser-114 and Arg-114 residues in sticks format and white and yellow element coloring, respectively; (**B**) same as (**A**), but with the DLA-88*001:04 α1α2 domain shown in surface format with element coloring; (**C**) same as (**B**) but with the DLA-88*508:01 Arg-114 sidechain shown in all-yellow spheres format, revealing how this residue decreases the space available in the E pocket.

**Figure 6 cells-12-01097-f006:**
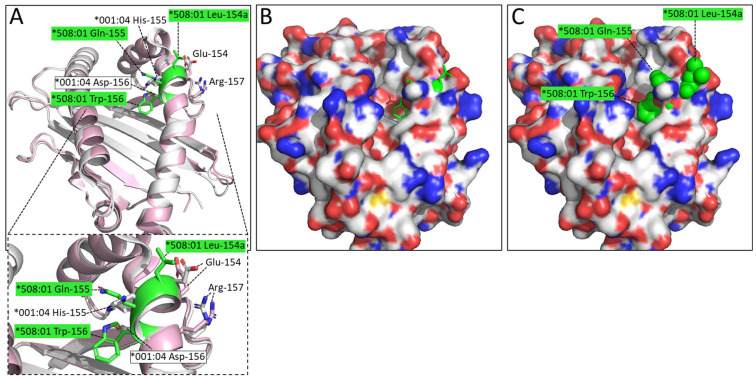
The LQW motif in pDLA-88*508:01 (PDB 5F1I) creates marked differences compared to pDLA-88*001:04 by having Leu-154a at the top of the α2 helix and Trp-156 at the border between the D and E pockets. Superposition of α1α2 domains: ((**A**), with zoomed-in view) pDLA-88*001:04 in white and pDLA-88*508:01 in pink, the sidechains of their positions 154-to-157 in sticks format and element coloring; the residues of the LQW motif are highlighted in green; (**B**) Same as (**A**) although in a slightly different orientation to show the open groove better, and with the DLA-88*001:04 α1α2 domain shown in surface format with element coloring and the sidechains of residues 154a-to-156 of DLA-88*508:01 154a-to-156 shown in all-green sticks format; (**C**) same as (**B**) but with the DLA-88*508:01 154a-to-156 residue sidechains shown in spheres format.

**Figure 7 cells-12-01097-f007:**
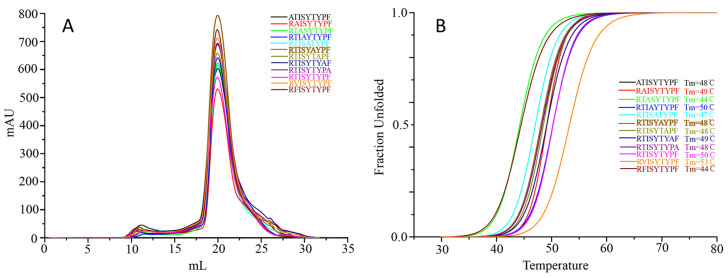
Peptide-induced assembly and pDLA-88*001:04 complex stability were tested using RTI9 peptide and several peptides derived thereof and in vitro refolding. (**A**) Gel filtration chromatograms of the refolded products. The main peaks represent the correctly refolded pDLA-88*001:04 complexes. (**B**) Thermal stabilities of the pDLA-88*001:04 complexes. Complexes of DLA-88*001:04/β2m bound to RTI9 or one of 11 mutant peptides were tested by CD spectroscopy. The denaturation curves of the complexes with the different peptides are indicated in different colors.

**Table 2 cells-12-01097-t002:** The interactions between peptide RTI9 and DLA-88*001:04 PBG.

Hydrogen Bonds and Salt Bridges				
Peptide		H Chain		
Residue	Atom	Residue	Atom	Van der Waals Forces
P1-R	O	Y159	OH	Y7, E63, Y159, T163, W167
	N	Y7	OH	
	N	Y171	OH	
P2-T	N	E63	OE1	Y7, Y9, M45, E63, T66, Y99, Y159
	N	E63	OE2	
	OG1	T66	OG1	
	OG1	E63	OE1	
	OG1	E63	OE2	
P3-I	N	Y99	OH	T66, Y99, H155, D156, Y159
P4-S	O	R73	NE2	R62, T66, E69
P5-Y				R73, H155
P6-T	OG1	R73	NE	T70, R73, Y74
	OE1	R73	NH2	
P7-Y	O	W147	NE1	W147, A150, G151, T152, H155
P8-P	O	W147	NE1	D77, K146
P9-F	OXT	Y84	OH	D77, T80, L81, Y84, R95, D116, Y123, I124, T143, W147
	OXT	T143	OG1	
	N	D77	OD1	

## Data Availability

The coordinates and structure factors of the DLA-88*001:04-CDV-H1RF9 complex have been deposited in the Protein Data Bank with the accession number 7CJQ.

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
