# Peer review of "Crystal Structure of a Classical MHC Class I Molecule in Dogs; Comparison of DLA-88*0 and DLA-88*5 Category Molecules"

_cells, 2023, doi:10.3390/cells12071097_

Round 1
Reviewer 1 Report
This is a very interesting manuscript, although dense, and not straightforward to read. I recommend publication. No major changes necessary.
Table I could add values after the decimal, as in the PDB
R factor (%)
25%
Rfree (%)
28%
- R-Value Free: 0.282
- R-Value Work: 0.253
Author Response
Dear Reviewer 1,
We are grateful for your time and effort to check our manuscript and your positive comments. We changed the values in Table 1 as requested.
Sincerely,
Also on behalf of the other authors,
Johannes M. Dijkstra
Reviewer 2 Report
The authors present the structure of the dog MHC-I complex, elucidating the structure characters of the peptide binding pocket. By comparing with structures of different alleles, the authors discuss a structural mechanism that separates allele DLA-88*0 and DLA-88*5 categories. The structure analysis of the complex is appropriate and the conclusion was discussed based on well-defined evidence. However, there are some places that need to be improved to facilitate the paper delivering the new information the study provided.
1. In the introduction, the significance of the different alleles' functions should be discussed more to show the importance of this study. And how this work can be related to the progress of the structural study of the human MHC-I complex.
2. The authors use anion-exchange chromatography to qualify the binding stability of the peptides (table S1), which is not directly relevant to the native environment. Another binding assay should be included to verify the binding affinity such as ITC or SPR.
3. In Figure 4, the residue of the pocket should also be colored properly to show different atoms instead of using a single purple color.
4. In Table S1, the sequence of the peptides should be aligned with the position of the residues since the authors were trying to compare the potential binding positions in the pocket (line 365).
5. In both Figures 5 and 6, zoomed-in views of A should be included to show the details of the comparison of the residues from 2 alleles.
Author Response
We thank Reviewer 2 for the time and effort to carefully check our manuscript, and for the compliment that our conclusion was based on well-defined evidence.
As to the individual comments (comments by the Reviewer in Italic):
1. In the introduction, the significance of the different alleles' functions should be discussed more to show the importance of this study. And how this work can be related to the progress of the structural study of the human MHC-I complex.
This is a brief report, which we assume will mainly be read by people who know about MHC. Furthermore, although the common models for functional explanation of MHC polymorphism are quite elegant, experimental evidence is scarce. However, we have now added the following new sentences to the Discussion section about (potential) functions of MHC polymorphism (lines 512-523):
"Balancing selection for maintaining MHC allelic variation is generally considered to be driven by different alleles presenting different peptides, thereby increasing the resistance of a population to pathogens that otherwise by mutating might more easily escape the immune protection in all individuals of a population (e.g., [53]). However, in teleost fish the MHC class I sequence information indicates that this cannot be the only reason, because in those species the conserved ancient allelic variation is not restricted to the PBG, and a function in allograft rejection has been proposed [54]. It was hypothesized that because of their mucosal skin and living in water, in teleost fish the allograft rejection against potential transferable cancers should be especially important [54], and the DLA-88*0 versus DLA-88*5 differences in canines may be an example how also in mammals allelic MHC variation can be increased to help allograft rejection of cancers that are transferred through mucosal tissues."
2. The authors use anion-exchange chromatography to qualify the binding stability of the peptides (table S1), which is not directly relevant to the native environment. Another binding assay should be included to verify the binding affinity such as ITC or SPR.
We realize that our method to study peptide binding is just one of several possible methods to study peptide binding and complex stability. However, as mentioned above, this is only a brief report study and we are not capable of doing additional experiments for this project.
3. In Figure 4, the residue of the pocket should also be colored properly to show different atoms instead of using a single purple color.
Changed as suggested.
4. In Table S1, the sequence of the peptides should be aligned with the position of the residues since the authors were trying to compare the potential binding positions in the pocket (line 365).
Changed as suggested.
5. In both Figures 5 and 6, zoomed-in views of A should be included to show the details of the comparison of the residues from 2 alleles.
Changed as suggested.
Reviewer 3 Report
This is an interesting paper.
I am not sure whether all readers will be familiar with some of the naming conventions used in this paper.
It would be helpful to remind readers that a “p” in front of a gene name refers to the protein.
Similarly, an explanation of what P1, P2, etc. means when referring to a peptide at the first use. And an introduction to the names of the pockets in the groove.
H chain should be explained also on first usage. Similarly PBG.
With regards to DLA nomenclature, I refer to the groups of DLA alleles as DLA-88*00x, and DLA-88*50x. Please change this throughout the paper.
Major issue:
In line 173, you state that there are 131 dog sequences and 9 wolf sequences.
Did you get this information from IPD?
If so, it is probably not correct, as there is an issue with IPD about which alleles are found in which species. The IPD database assumes an allele can only be in one species, so in order to have an allele listed in several different species, it has to be entered multiple times. This has not been done.
Similarly, IPD does not recognise “canis lupus familiaris” i.e. the domestic dog, as separate from “canis lupus”, the wolf.
If you have another source of this information, please indicate your source in the manuscript.
In general, it is unwise to state DLA alleles are “only dog” or “only wolf”, since many are shared, and found in both dogs and wolves (and other canids too).
Therefore please remove the “W”s from Figure 1.
Minor comments/corrections:
Line 58: do you mean DLA-79?
Line 62 you say there are 140 sequences, but on line 85 you then sat there are 142 sequences. Which is it?............ mention here the extra two CTVT ones have been added
Line 171: plural of wolf is wolves
Line 483: plural of wolf is wolves
Line 237: I do not think these two alleles align properly with DLA-88……
Line 254: Could you explain how you came to the conclusion for the location of the insertion in exon 3.
Fig 2 has the amino acids as ……TAE-HDRNY……..,
whereas other papers suggest ……TAEHD-RNY………
Line 461: “:presence and absence…” i.e. not plural
Line 502: do you mean ligand? Not ligan?
Figure 1: would it be worth indicating which alleles are DLA-88L?
Table S1: in the footnote you refer to DLA-88*001:01, should this be DLA-88*001:04?
Author Response
We thank Reviewer 3 for the careful and detailed checking, and for the compliment that our article is interesting. The comments have helped to improve the manuscript, including the removing of some silly typos. Thank you!
Our responses to the individual comments are (the Reviewer's comments are in Italic):
It would be helpful to remind readers that a “p” in front of a gene name refers to the protein.
Our response: The Reviewer is right and we should explain terminology better to the reader.
It now says (lines 287-290): "The X-ray data collection statistics of this structure—which for simplicity is named pDLA-88*001:04 (the “p” refers to being a complex of heavy chain, β2m, and peptide) in most of this study—are summarized in Table 1."
Similarly, an explanation of what P1, P2, etc. means when referring to a peptide at the first use. And an introduction to the names of the pockets in the groove.
We have now added information on the meaning of "P" to the Introduction (lines 69-71): "Amino acid substitution analysis revealed that anchor residues for binding DLA-88*001:04 involve peptide ligand positions P2 (“P” for peptide ligand position) and P9."
However, we believe that our descriptions of the pocket names is already well enough described in the sentence (lines 318-320): "Based on the pockets in the peptide binding groove (PBG) of human pHLA-A2 [40], PBGs of various pMHC-Is have been divided into pockets A-to-F regions, as we do for pDLA-88*001:04 in the present study (Fig. 4)."
H chain should be explained also on first usage. Similarly PBG.
We have now added, to the Results section (lines 292-296 and 318-320):
"In pDLA-88*001:04, the polar interactions between heavy (H) chain and β2m include the highly conserved pMHC-I-typical interactions between β2m Trp-59 and H chain Gln-96 and Asp-119, between β2m Asp-52 and H chain Arg-48, and between β2m Tyr-10 and H chain Pro-235 (Fig. 3C) [36,40,41]."
and
"Based on the pockets in the peptide binding groove (PBG) of human pHLA-A2 [40], PBGs of various pMHC-Is have been divided into pockets A-to-F regions, as we do for pDLA-88*001:04 in the present study (Fig. 4)."
With regards to DLA nomenclature, I refer to the groups of DLA alleles as DLA-88*00x, and DLA-88*50x. Please change this throughout the paper.
We apologize, but prefer the "DLA-88*0 and DLA-88*5 categories" nomenclature, and consider this to be fully consistent and not in conflict with your nomenclature. The names DLA-88*00x and DLA-88*50x refer to the allele naming system, but as category name have the problem that some alleles in this category do not have a zero at the second number position in their name.
Major issue:
In line 173, you state that there are 131 dog sequences and 9 wolf sequences.
Did you get this information from IPD?
We retrieved all the DLA-88 sequences from IPD and compared them with DLA-88 sequences retrieved from UniProt, for which it was easier to find the origin of the sequences. However, based on the Reviewer's comments, we have now scrutinized the data better, and found that three of the nine DLA-88 alleles reported for wolf have also been reported for dog. This has now been specified in the Fig. 1 legend (lines 237-238): "The DLA-88 alleles *024:02 ([37] and GenBank LR890075), *039:01 [37,38], and *046:01 (GenBank AKS10581 and [38]) were reported for both dog and gray wolf."
If so, it is probably not correct, as there is an issue with IPD about which alleles are found in which species. The IPD database assumes an allele can only be in one species, so in order to have an allele listed in several different species, it has to be entered multiple times. This has not been done.
Similarly, IPD does not recognise “canis lupus familiaris” i.e. the domestic dog, as separate from “canis lupus”, the wolf.
If you have another source of this information, please indicate your source in the manuscript.
You are probably correct, as these sub-species can produce fertile offspring. We have now added such information to the materials and methods section (lines 75-85):
"Sequences were retrieved from the MHC dataset of the Immuno-Polymorphism Database (IPD) [21] (https://www.ebi.ac.uk/ipd/mhc/, accessed in December 2022), the National Center for Biotechnology Information (NCBI) (https://pubmed.ncbi.nlm.nih.gov, accessed in December 2022), and UniProt (https://www.uniprot.org/, accessed in December 2022). UniProt provided information on whether the sequences were found in dog or gray wolf, and provided links to NCBI accessions with detailed information. For DLA-88 alleles found in wolf, we tried to find their reporting in dogs by BLASTP similarity searches in the NCBI database and by checking literature. Except for two DLA-88 alleles in CTVT cancer cells, only sequences that were (also) deposited in the IPD database—which we interpreted as a quality mark by the dog MHC research community—were analyzed in the present study."
In general, it is unwise to state DLA alleles are “only dog” or “only wolf”, since many are shared, and found in both dogs and wolves (and other canids too).
Therefore please remove the “W”s from Figure 1.
We understand the concern by the Reviewer. We have now changed to (lines 181-184): "The a1 + a2 domain sequences of all the 140 DLA-88 alleles available in the IPD-MHC database were compared. Of these, to the best of our knowledge, 131 were only reported for dog (Canis lupus familiaris), six only for gray wolf (Canis lupus lupus), and three for both subspecies, but further investigations will probably find that more alleles are shared."
In the Fig. 2 legend, we now explain W as "(also in) Wolf"
For our message of balancing selection, it is important that despite the few alleles found in wolf, they also contain both DLA-88*0 and DLA-88*5 category alleles. Therefore, we like to keep the "W" indications for wolf in Figs. 1 and 2, but hope that together with the main text the reader can now better understand the situation.
Minor comments/corrections:
Line 58: do you mean DLA-79?
Thank you for noticing the error.
Line 62 you say there are 140 sequences, but on line 85 you then sat there are 142 sequences. Which is it?............ mention here the extra two CTVT ones have been added
We have now modified to (lines 93-94) "The analysis involved 142 amino acid sequences (140 from the IPD database and two reported for CTVT)."
Line 171: plural of wolf is wolves
Line 483: plural of wolf is wolves
Our apologies. Thank you for noticing.
Line 237: I do not think these two alleles align properly with DLA-88……
In Fig. 2, their alignment with other DLA-88 alleles is shown, which goes perfectly well. If you would consider them to somehow not have a full DLA-88 function, that may very well be possible because within these cancer cells they are under different evolutionary pressures than in the organism.
Line 254: Could you explain how you came to the conclusion for the location of the insertion in exon 3.
Fig 2 has the amino acids as ……TAE-HDRNY……..,
whereas other papers suggest ……TAEHD-RNY………
We have now added the following information to the Materials and Methods section (lines 85-87; you can also see it in Fig. 6): "Sequences were aligned by hand [22], which in comparison to computerized alignments corrects and homogenizes the position of the gap in the a2 domain; this position agrees with the structural comparison between pDLA-88*001:04 and pDLA-88*508:01."
Line 461: “:presence and absence…” i.e. not plural
Changed as suggested.
Line 502: do you mean ligand? Not ligan?
Our apolologies.
Figure 1: would it be worth indicating which alleles are DLA-88L?
Maybe, but we feel that this debate should be left to the dog MHC sequence specialists. We now have just added a sentence to the Fig. 1 figure legend, only showing the principle (lines 235-237): "A few of the here compared sequences of the DLA-88*0 category may be expressed from the DLA-88L/DLA-12 locus [37]."
Table S1: in the footnote you refer to DLA-88*001:01, should this be DLA-88*001:04?